

# Critical multi-stranded approach for determining the ecological values of diatoms in unique aquatic ecosystems of anthropogenic origin

Rafał M. Olszyński*, Ewelina Szczepocka and
Joanna Żelazna-Wieczorek*

University of Lodz, Faculty of Biology and Environmental Protection, Depertment of Algology
and Mycology, Lodz, Poland
* These authors contributed equally to this work.

## ABSTRACT

**Background:** The ecological state of surface waters is typically assessed by a multi-aspect approach based on a determination of its chemical and physical parameters, by hydromorphology and the use of indicator organisms such as benthic diatoms. By assigning ecological indicator values, it is possible to create diatom indices which serve as the basic tool in assessing the ecological status of surface waters. These ecological indicator values are set according to classification systems, such as the *Van Dam, Mertens & Sinkeldam (1994)* system, which classifies species of diatoms according to seven different ecological factors. However, recent studies on the autecology of diatoms have shown the need to verify and establish new ecological indicator values. To this end, aquatic ecosystems are good environments to observe the range of tolerance of benthic diatoms to environmental conditions due to their unique physical and chemical parameters. The aim of the present study was to propose the establishment of new, or altered, ecological indicator values, according to the *Van Dam, Mertens & Sinkeldam (1994)* classification, of species of diatoms characteristic of three post-mining aquatic ecosystems.
**Methods:** In total, 36 species were identified that were characteristic of three waterbodies: a salt aquatic complex (water outflow, a drainage ditch and a pond), mined iron ore reservoirs and a mined lignite reservoir. Their ecological indicator values were specified using OMNIDIA software, and the environmental conditions prevailing in the studied ecosystems were determined. Of the 36 characteristic species, 16 lacking at least one assigned ecological indicator value were analyzed further. The analysis identified three groups of selected characteristic species which showed a correlation, or lack of such, to the tested physical and chemical parameters.
**Results:** Based on this multistage study of the autecology of characteristic diatoms, comprising an analysis of environmental conditions, literature analysis and reference ecological indicator values of other species, it is proposed that 32 ecological indicator values be established or adjusted for 16 species, and that *Planothidium frequentissimum* be excluded from water quality assessments.

Corresponding authors
Rafał M. Olszyński,
rafal.olszynski@biol.uni.lodz.pl
Joanna Żelazna-Wieczorek,
joanna.zelazna@biol.uni.lodz.pl

## INTRODUCTION

Diatoms (Bacillariophyta) are one of the main biotic elements used in the biological assessment of the ecological state of surface waters (Water Framework Directive, *European Union, 2000*). Due to the fact that many countries are obliged to continually engage in biomonitoring, there is a clear need to develop flawlessly functioning methods based on the standardized use of diatoms as bioindicators (*Kahlert et al., 2016*; *Poikane, Kelly & Cantonati, 2016*; *Szczepocka & Żelazna-Wieczorek, 2018*). Diatom indices and ecological systems based on the bioindication values of particular diatom species, derived from various environmental parameters, constitute a fundamental tool in the biological assessment of environments. Diatom indices have been commonly used to assess flowing and standing water for over 20 years (*Kelly et al., 2008*; *Harding & Taylor, 2014*; *Szczepocka et al., 2014*; *Hutorowicz & Pasztalenic, 2014*; *Holmes & Taylor, 2015*; *Żelazna-Wieczorek & Nowicka-Krawczyk, 2015*; *Kolada et al., 2016*).

Currently, many countries use the OMNIDIA program (*Lecointe, Coste & Prygiel, 1993*) as a biological assessment tool. Its latest version (version 6.0.6) allows the calculation of 18 diatom indices, and the determination of seven environmental parameters for eight ecological systems. However, the specific ecological indicator values of many of the species given in the OMNIDIA database are absent or have not been updated in response to recent research. To complete these missing values, and to verify existing ones, further studies are needed of the ecological optima and tolerance of diatom species in different types of aquatic ecosystems.

Due to their specific environmental conditions, post-mining reservoirs represent an extremely valuable source of information for the study of ecological diatom tolerance ranges. Some studies of these environments have been performed, but these have addressed diatom paleoecology and their role as indicators of past climatic or environmental change (*De Haan et al., 1993*; *Rakowska, 1996*; *Thomas & John, 2006*; *Sienkiewicz & Gąsiorowski, 2016*). Until now, the autecology of diatoms in post-mining reservoirs has rarely been studied (*Van Landingham, 1968*; *De Haan et al., 1993*; *Rakowska, 1996*; *Ferreira da Silva et al., 2009*; *Luís et al., 2009*, *2016*; *Sienkiewicz & Gąsiorowski, 2016*).

The present study examines the diatom assemblages present in three post-mining reservoirs of various geological origins. Due to variations in their environmental parameters, these bodies of water serve as specific and unique habitats for the development of these algae. The diatom assemblages quickly adapt to the currently prevailing conditions, which is manifested in the presence of taxa characteristic of these specific parameters. Considering their large share of the assemblage, the index values of the assemblages constitute the most important component in the calculation of diatom indices. These species are therefore of the greatest importance for surface water biomonitoring.

The aim of the present study was to identify the species of diatoms characteristic of the three studied types of post-mine reservoirs. Following this, taxa that did not have at least one ecological indicator value specified in the OMNIDIA database, according to the environmental parameters given by *Van Dam, Mertens & Sinkeldam (1994)*, were

identified. New ecological indicator values were proposed based on the relationship between the occurrence of the individual species and certain selected physical and chemical parameters, or existing ones were verified.

The *Van Dam, Mertens & Sinkeldam (1994)* ecological system is one of the main systems on which the OMNIDIA program is based. It describes the ecological indicator values of diatoms according to pH, salinity, nitrogen uptake metabolism, oxygen requirement, saprobity, trophic state and moisture aerophily. These values play a key role in calculating diatom indices, and hence need to be kept up to date to enable accurate routine biomonitoring.

## MATERIALS AND METHODS

### Study area

The study was performed on three waterbodies created through exploration for mineral deposits or were formed by the closure of mines. All three are located in the Łódzkie and Wielkopolskie voivodeships, Central Poland.

The first complex of waterbodies—Pełczyska (PE), is situated in the village of Pełczyska, between Łódź and Łęczyca (Łódzkie voivodeship) (Fig. 1). As the local area is characterized by the presence of salt deposits, numerous wells were sunk in the eighteenth century to obtain brine. Currently, salt water flows out of one of them. This area has been studied by biologists and hydrobiologists since the 1960s (*Olaczek, 1963*; *Pliński, 1966*, *1969*, *1971a*, *1971b*, *1971c*, *1973*; *Żelazna-Wieczorek, 1996*, *2002*; *Żelazna-Wieczorek, Olszyński & Nowicka-Krawczyk, 2015*; *Żelazna-Wieczorek & Olszyński, 2016*). The waterbodies chosen for our research form the PE hydrological complex located in the vicinity of farmland; it comprises the salt water outflow, a drainage ditch and a pond, which acts as the receiver of the water.

The second complex of waterbodies—Łęczyca (LE), urban reservoirs located within the city of Łęczyca (Łódzkie voivodeship) (Fig. 1). The reservoirs were created following the flooding the open-cast iron ore mine in the 1990s. This area is rich in syderite deposits, which are accompanied by other minerals. The complex consists of three connected reservoirs: two are directly connected to each other (LEP1 and LEP2), and the third (LEP3) is connected to LEP2 via a water drainage ditch (*Olszyński & Żelazna-Wieczorek, 2018*). All three are located in an area with houses, garden plots and partly-wooded areas.

The third waterbody—Bogdałów reservoir (BO), created by the flooding of an opencast brown coal mine. It is located in the village of Bogdałów (Wielkopolskie voivodeship) in an area rich in lignite deposits (Fig. 1). Lignite from quaternary deposits was exploited since 1977 until 1991 to a depth of 50 m. Due to the specific construction of the open-pit area, being characterized by the thickest layer of poorly permeable boulder clay in the region. This pit was later transformed into a storage site for quarried rocks in Koźmin. Finally, in 1993/1994, the drainage and runoff of surface waters were blocked to form a reservoir with a depth of about 12 m surrounded by forest (*Gabryś-Godlewska et al., 2004*; *Gadomska et al., 2007*; *Orlikowski & Szwed, 2009*; *Kasztelewicz, 2011*).

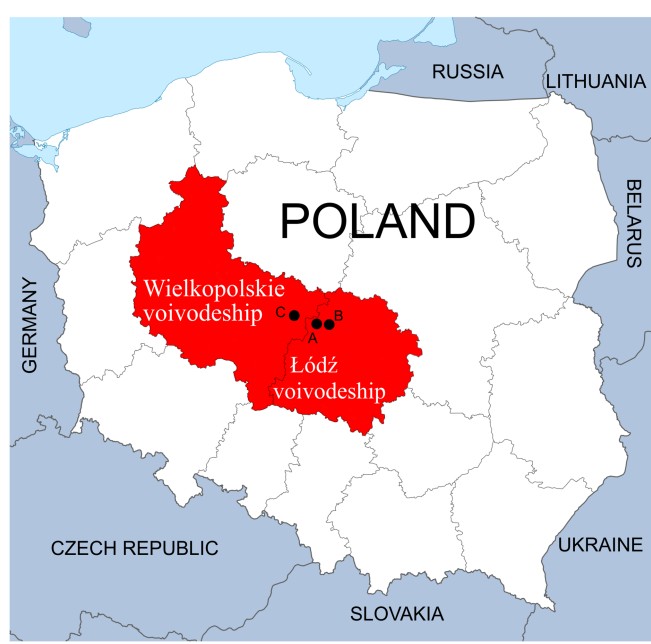

**Figure 1 Location of sampling points in the Łódzkie and Wielkopolskie voivodships, Poland.**
(A) Pełczyska (PE). (B) Łęczyca (LE). (C) Bogdałów (BO).

## Samples

Samples of benthic diatoms from sediments and water samples were collected quarterly
(once in any season) from each hydrological waterbodies. Analysis of all water samples
($Ca^{2+}$, $Mg^{2+}$, $Na^+$, $K^+$, $Fe^{2+/3+}$, $Mn^{3+}$) involved flame absorption spectrophotometry
SpectrAA 300 (Varian, Palo Alto, CA, USA) (detection limit is 0.05 mg/L) and UV-vis
spectrophotometry S.330 (Marcel, Poland) in the case of $NH_4^+$ (d.l. is 0.001 mg/L)
and $PO_4^{3-}$ (d.l. is 0.01 mg/L). $SO_4^{2-}$ was determined by the gravimetric method
(PN-C-04566-09), $Cl^-$ by Mohr's method (PN-ISO 9297). The chemical analyses
were performed in the Laboratory of the Department of Geology at the Faculty of
Geographical Sciences, University of Lodz and the Laboratory of Computer and
Analytical Techniques at the Faculty of Biology and Environmental Protection,
University of Lodz. The water temperature, pH and electric conductivity were measured
in situ (Elmetron CP-401 and CC-401 devices). The following sampling points were
established:

Pełczyska (51°58′34.47″N; 19°14′21.11″E)—outflow (D.PESB) (4 samples, both water
and benthic), ditch (D.PEDB) (one water and four benthic samples) and pond (D.PEPB)
(4 samples, both water and benthic); samples were collected quarterly from July 2013
to March 2014;

Łęczyca (52°3′5.30″N; 19°11′50.24″E)—reservoir 1 (D.LEP1), reservoir 2 (D.LEP2) and
reservoir 3 (D.LEP3), samples were collected quarterly from March 2014 to December
2015 (six water and eight benthic samples from each reservoirs);

Bogdałów (52°2′51.29″N; 18°35′51.49″E)—reservoir (D.BOZB), samples were collected
quarterly from March 2015 to December 2016 (eight samples, both water and benthic).

In total, 44 benthic samples were collected. The permanent slides were prepared according to *Żelazna-Wieczorek (2011)*. To obtain pure diatom frustule the material was chemically treated using a $H_2SO_4$ and $H_2Cr_2O_7$. The cleaned diatom precipitate was mounted on permanent slides using Naphrax® synthetic resin.

Qualitative and quantitative analysis of diatoms was performed using a Nikon Eclipse 50i light microscope (LM) under 1000× magnification (plan oil-immersion objective 100×/1.25): the diatoms were identified and counted for up to 500 valves in each permanent slide. Light photomicrographs were taken with an OPTA-TECH digital camera.

When diatoms were difficult to identify using LM they were subjected to scanning electron microscope (SEM) analysis using a Phenom ProX (gold layer of 8 and 20 nm, at 10 kV, low vacuum mode) at the Laboratory of Microscopy Imaging and Specialist Biological Techniques, Faculty of Biology and Environmental Protection, University of Lodz.

## Data processing and statistical analysis

The average percentage (AP) for a given species was determined based on the percentage contribution (%) of the species in the samples tested for a given hydrological object (*Żelazna-Wieczorek, 2011*). Species whit AP ≥5% for each hydrological object were identified as dominant.

The incidence was determined according to the *Tümpling & Friedrich (1999)* coefficient according to the range values: 100–75% euconstant taxa (EC), 75–50% constant taxa (CN), 50–25% accessory taxa (AC) and 25–1% accidental taxa (AD) (*Tümpling & Friedrich, 1999*).

Multidimensional scaling analysis (nMDS) based on Bray–Curtis similarity coefficients was used to identify natural groupings of samples. The results are given as a 3D diagram in which the degree of similarity is represented as the distances between particular points (samples), with greater distances indicating a lower degree of similarity. The reliability of the ordering of the assemblage is represented by the stress value, which reflects how well the ordination summarizes the observed distances among the samples. A three-dimensional presentation, whose stress value is lower, is likely to be more satisfactory than a two-dimensional one (*Clarke & Gorley, 2015*).

The Shade Plot analysis, based on the Bray–Curtis similarity coefficient, was used to identify the diatom species that have the strongest influence on the similarities between the samples demonstrated in the nMDS analysis. Shade Plot analysis compares two data matrices with each other and then groups them on two levels, according to the similarity of the samples and the factors affecting their similarity, that is, diatom species. The results are represented graphically by shading individual cells: the intensity of the shading indicates the degree of the influence of a given factor (species) on the position of its sample within a given similarity cluster. The range of the shading was determined on the basis of $\log(x + 1)$ ($x$—number of valves).

The SIMPER analysis was used to determine the characteristic species distinguishing the studied ecosystems. This method examines the participation of each variable in the overall similarity between groups of samples, thus indicating the species with the greatest influence on the degree of similarity, or dissimilarity, between particular samples and hydrological objects. This analysis is also based on the Bray–Curtis similarity coefficient;

however, unlike the nMDS method, in which one trial is compared to all the other samples, the SIMPER analysis compares a single sample to each subsequent sample (*Żelazna-Wieczorek, 2011*). The results indicate the species which most strongly differentiated a sampled site from the others, and to what extent. A species was regarded as being characteristic of the studied ecosystem if it was characterized by a mean dissimilarity ≥2 according to the SIMPER analysis, and a higher mean abundance greater in one ecosystem than the other.

In total, 19 physical and chemical parameters of water were measured in the studied ecosystems. The results of the correlation analysis found 15 physical and chemical parameters indicating an environmental conditions. The parameters were subjected to principal component analysis (PCA) to determine which had the strongest effect on the selected species.

Using the information from the OMNIDIA database, the environmental conditions for each sampling point were determined according to *Van Dam, Mertens & Sinkeldam (1994)* (Table S1). Following this, the percentage share of diatom species included in each ecological indicator value class was indicated. For species found to be characteristic of the studied ecosystems, classes of ecological indicator values were assembled. Taxa which had at least one value of 0 (unknown) were selected for further analysis.

The selected ecological indicator values according to *Van Dam, Mertens & Sinkeldam (1994)* were verified, or new ones established, for the species found to be characteristic of the studied ecosystems according to three premises: previous literature reports about ecological indicator values of those species, chemical and physical conditions analysis, and the classification of the environmental conditions according to *Van Dam, Mertens & Sinkeldam (1994)*.

The analyses were performed using PRIMER 7.0.13 (nMDS, Shade Plot, SIMPER), OMNIDIA 6.0.6 and STATISTICA 13 (PCA), software.

## RESULTS

### Chemical analysis of water samples

The mean values and range of all tested parameters are given in Table 1.

The PE hydrological complex was characterized by elevated values of electric conductivity, reaching as high as 9230 μS cm$^{-1}$. The pH changed with the direction of water outflow: a slightly acidic reaction was observed in the outflow and an alkaline one in the pond. Due to the geological profile of the region, the water flowing out of the well contained a high concentration of chloride ions, whose gradient decreased with the flow of water through the ditch to the pond. In addition, higher concentrations of the cations $Mg^{2+}$, $Ca^{2+}$, $Na^+$ and $K^+$ were observed compared to other ecosystems, as well as the anions $HCO_3^-$, $PO_4^{3-}$ and $SO_4^{2-}$.

The $K^+$ concentration is acknowledged parameter coming from agricultural activity, in particular animal husbandry, or municipal wastes (*Macioszczyk & Dobrzyński, 2002*).

Each of the sampling points in the PE complex was characterized by different chemical parameters, resulting in differences between the habitats. The highest electrolytic conductivity was noted in the outflow, which was mainly influenced by the concentrations

**Table 1 Physical and chemical parameters in the examined sampling sites.** The minimum, maximum and mean values.

| | Pełczyska (PE) | | | | | | | | | Łeczyca (LE) | | | | | | | | | Bogdałów (BO) | | |
|---|---|---|---|---|---|---|---|---|---|---|---|---|---|---|---|---|---|---|---|---|---|
| | Outflow (D.PESB) | | | Ditch (D.PEDB) | | | Pond (D.PEPB) | | | L1 (D.LEP1) | | | L2 (D.LEP2) | | | L3 (D.LEP3) | | | Reservoir (D.BOZB) | | |
| | min | max | avar. | min | max | avar. | min | max | avar. | min | max | avar. | min | max | avar. | min | max | avar. | min | max | avar. |
| pH | 6.6 | 7.7 | 7.0 | 6.4 | 6.4 | 6.4 | 7.8 | 9.6 | 8.7 | 7.9 | 8.6 | 8.2 | 7.5 | 8.6 | 8.2 | 7.5 | 8.8 | 8.3 | 7.9 | 8.3 | 8.1 |
| Conductivity ($\mu S\ cm^{-1}$) | 4,450 | 9,230 | 6,699 | 5,170 | 5,170 | 5,170 | 2,645 | 5,150 | 3,646 | 657 | 865 | 743 | 558 | 836 | 728 | 472 | 778 | 680 | 505 | 734 | 623 |
| T (°C) | 6.8 | 13.9 | 9.1 | 7.8 | 7.8 | 7.8 | 1.1 | 17.3 | 7.1 | 1.6 | 20.6 | 11.1 | 0.6 | 21.1 | 11.1 | 2.2 | 15.2 | 9.2 | 4.7 | 22.7 | 11.4 |
| $HCO_3^-$ ($mg\ L^{-1}$) | 345 | 744 | 475 | 610 | 610 | 610 | 284 | 451 | 352 | 211 | 339 | 260 | 168 | 275 | 210 | 183 | 290 | 241 | 174 | 369 | 270 |
| $CO_2^{(HCO3-)}$ ($mg\ L^{-1}$) | 124 | 268 | 171 | 220 | 220 | 220 | 102 | 163 | 126 | 76 | 122 | 94 | 61 | 99 | 75 | 66 | 105 | 87 | 63 | 133 | 96 |
| $Cl^-$ ($mg\ L^{-1}$) | 1,585 | 2,976 | 2,426 | 1,006 | 1,006 | 1,006 | 685 | 1,524 | 1,053 | 57 | 93 | 78 | 60 | 93 | 76 | 67 | 106 | 78 | 41 | 52 | 45 |
| $N_{NH4}$ ($mg\ L^{-1}$) | 0.02 | 1.63 | 0.45 | 0.15 | 0.15 | 0.15 | 0.00 | 0.36 | 0.11 | 0.02 | 0.71 | 0.22 | 0.03 | 0.53 | 0.26 | 0.00 | 1.16 | 0.32 | 0.00 | 0.07 | 0.02 |
| $NH_4^+$ ($mg\ L^{-1}$]) | 0.03 | 2.09 | 0.71 | 0.19 | 0.19 | 0.19 | 0.00 | 0.46 | 0.14 | 0.03 | 0.55 | 0.22 | 0.03 | 0.54 | 0.28 | 0.00 | 0.90 | 0.31 | 0.00 | 0.09 | 0.03 |
| $PO_4^{3-}$ ($mg\ L^{-1}$) | 0.60 | 12.46 | 3.73 | 8.57 | 8.57 | 8.57 | 0.85 | 9.10 | 3.99 | 0.22 | 0.53 | 0.36 | 0.05 | 0.57 | 0.41 | 0.09 | 0.57 | 0.34 | 0.27 | 0.45 | 0.34 |
| $P_{PO4}$ ($mg\ L^{-1}$) | 0.20 | 4.11 | 1.23 | 2.83 | 2.83 | 2.83 | 0.28 | 3.00 | 1.32 | 0.07 | 0.17 | 0.12 | 0.02 | 0.19 | 0.13 | 0.03 | 0.19 | 0.11 | 0.09 | 0.15 | 0.12 |
| $SO_4^{2-}$ ($mg\ L^{-1}$) | 176 | 198 | 188 | 165 | 165 | 165 | 151 | 197 | 176 | 71 | 147 | 107 | 67 | 122 | 101 | 63 | 91 | 75 | 110 | 147 | 123 |
| $S_{SO4}$ ($mg\ L^{-1}$) | 58.8 | 66.2 | 62.9 | 55.1 | 55.1 | 55.1 | 50.3 | 65.7 | 58.7 | 23.5 | 49.0 | 35.8 | 22.3 | 40.8 | 33.8 | 21.0 | 30.3 | 25.1 | 37.9 | 49.2 | 41.5 |
| COLOR ($mgPt\ dm^{-3}$) | 25 | 160 | 81 | 140 | 140 | 140 | 50 | 120 | 78 | 10 | 60 | 27 | 9 | 60 | 27 | 12 | 60 | 31 | 4 | 10 | 6 |
| $Mn^{3+}$ ($mg\ L^{-1}$) | 0.17 | 0.51 | 0.36 | 0.14 | 0.14 | 0.14 | 0.03 | 0.25 | 0.13 | 0.00 | 0.05 | 0.02 | 0.01 | 0.03 | 0.02 | 0.01 | 0.07 | 0.03 | 0.00 | 0.01 | 0.00 |
| $Fe^{2+/3+}$ ($mg\ L^{-1}$) | 0.25 | 0.41 | 0.32 | 0.25 | 0.25 | 0.25 | 0.05 | 0.24 | 0.11 | 0.03 | 0.27 | 0.09 | 0.00 | 0.13 | 0.04 | 0.01 | 0.07 | 0.03 | 0.01 | 0.09 | 0.02 |
| $Mg^{2+}$ ($mg\ L^{-1}$) | 39.5 | 48.7 | 45.0 | 35.9 | 35.9 | 35.9 | 25.6 | 37.0 | 32.1 | 11.6 | 19.1 | 15.2 | 10.3 | 18.3 | 15.1 | 9.5 | 15.3 | 12.5 | 9.6 | 15.7 | 11.7 |
| $Ca^{2+}$ ($mg\ L^{-1}$) | 171.7 | 216.2 | 195.2 | 165.0 | 165.0 | 165.0 | 75.8 | 139.8 | 121.1 | 58.0 | 143.6 | 89.0 | 59.5 | 117.6 | 79.1 | 52.6 | 86.4 | 65.5 | 77.7 | 117.6 | 86.3 |
| $Na^+$ ($mg\ L^{-1}$) | 500.7 | 1537.4 | 1227.1 | 453.3 | 453.3 | 453.3 | 277.3 | 681.8 | 455.9 | 15.6 | 40.4 | 30.9 | 20.3 | 42.4 | 28.3 | 22.0 | 42.4 | 33.0 | 24.0 | 43.1 | 32.6 |
| $K^+$ ($mg\ L^{-1}$) | 8.5 | 124.8 | 42.4 | 109.8 | 109.8 | 109.8 | 58.6 | 68.8 | 63.5 | 4.5 | 9.5 | 7.2 | 5.1 | 9.3 | 7.3 | 5.6 | 10.7 | 8.2 | 0.1 | 3.6 | 1.5 |

of $Cl^-$, $Na^+$ and $HCO_3^-$ ions. The maximum concentration of $HCO_3^-$ ions was recorded in Pełczyska outflow in March 2014 (D.PESB.250314); in the other locations, it did not exceed 410 mg $L^{-1}$.

Low concentrations of $K^+$ ions were observed throughout the entire studied PE complex; however, maximum values were recorded in the locations characterized by the highest $HCO_3^-$ ion content. The highest concentration of $Ca^+$ ions of all ecosystems was recorded in the outflow. The ditch represented an intermediate section between the PE sampling points. However, as it is susceptible to periodic drying, limited chemical data was collected from this habitat and hence it was not possible to assess its chemical and physical nature.

The lowest electrolytic conductivity was found in the pond, which displayed lower concentrations of $Cl^-$, $Na^+$ and, to a lesser degree, $HCO_3^-$. The pH of the water never dropped below 8, except in one case in March 2014. In the pond, the concentration of $K^+$ remained relatively unchanged, which could be related to the fact that the reservoir was also a receiver of waters flowing from the surrounding arable fields. The pond was also characterized by the lowest concentration of $Ca^{2+}$ and $Mg^{2+}$. In the summer periods, a significant reduction in the water table level and occasional drying of the reservoir were noted.

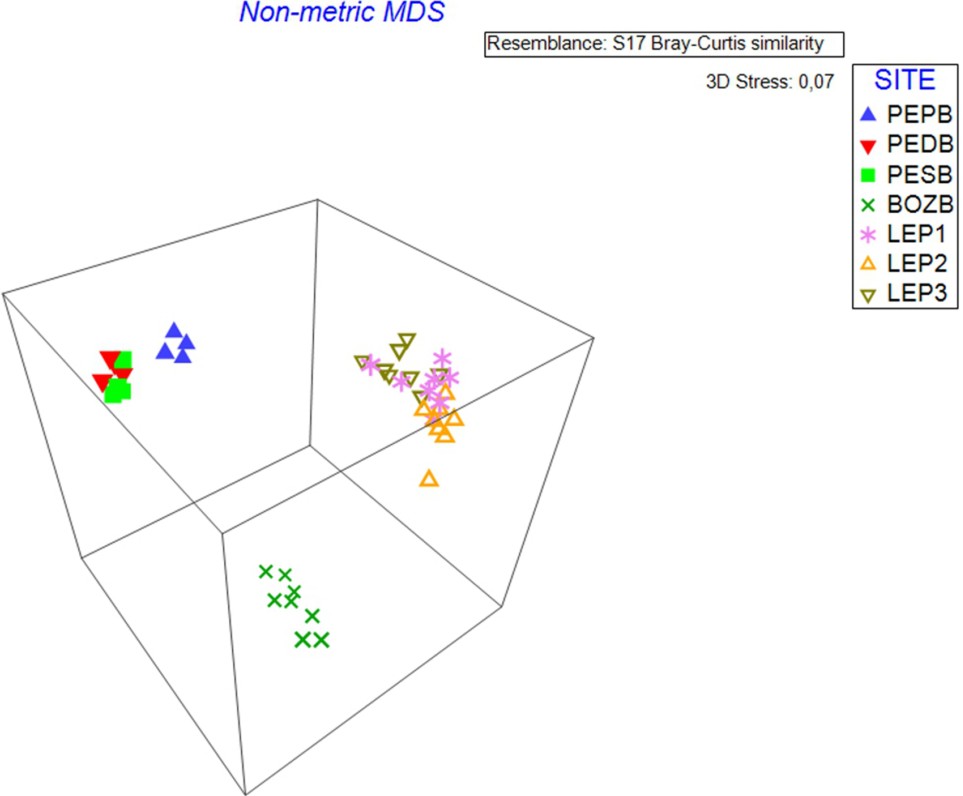

**Figure 2 nMDS 3D analysis.** The diagram shows three distinct clouds of samples which coincide with the three hydrological objects.

The urban reservoirs in Łęczyca (LE) were characterized by a slightly alkaline water reaction, which was similar in all reservoirs during the course of the study. No elevated concentrations of $Fe^{2+/3+}$ and $Mn^{3+}$ ions were observed. The content of $SO_4^{2-}$ anions was not higher than in other waterbodies studied. The concentration of $HCO_3^-$ ions was lower than that observed in BO and PE. No significant differences in chemical and physical parameters were observed between the individual sampling points constituting LE.

The Bogdałów (BO) reservoir was characterized by an alkaline reaction. It's $K^+$, $Cl^-$ and $NH^{4+}$ ion content was the lowest of the studied ecosystems.

### Diatom samples

A total of 381 diatom taxa were identified in 44 benthic samples: 139 in PE, 192 in LE and 188 in BO. The dominant species in PE were *Navicula veneta*, and *Nitzschia frustulum*, in LE *Cyclostephanos dubius* and *Stephanodiscus hantzschii*, in BO *Achnanthidium minutissimum*, *Pantocsekiella ocellata* and *Mastogloia smithii*. In the examined ecosystems, the most commonly identified classes were accidental (PE-84; LE-111; BO-86), accessory (PE-25; LE-35) and euconstant taxa (BO-39) (Fig. S1).

### nMDS analysis

nMDS analysis (stress level = 0.07) identified the variation between samples for each studied hydrological object (Fig. 2). The samples taken from BO constitute a separate

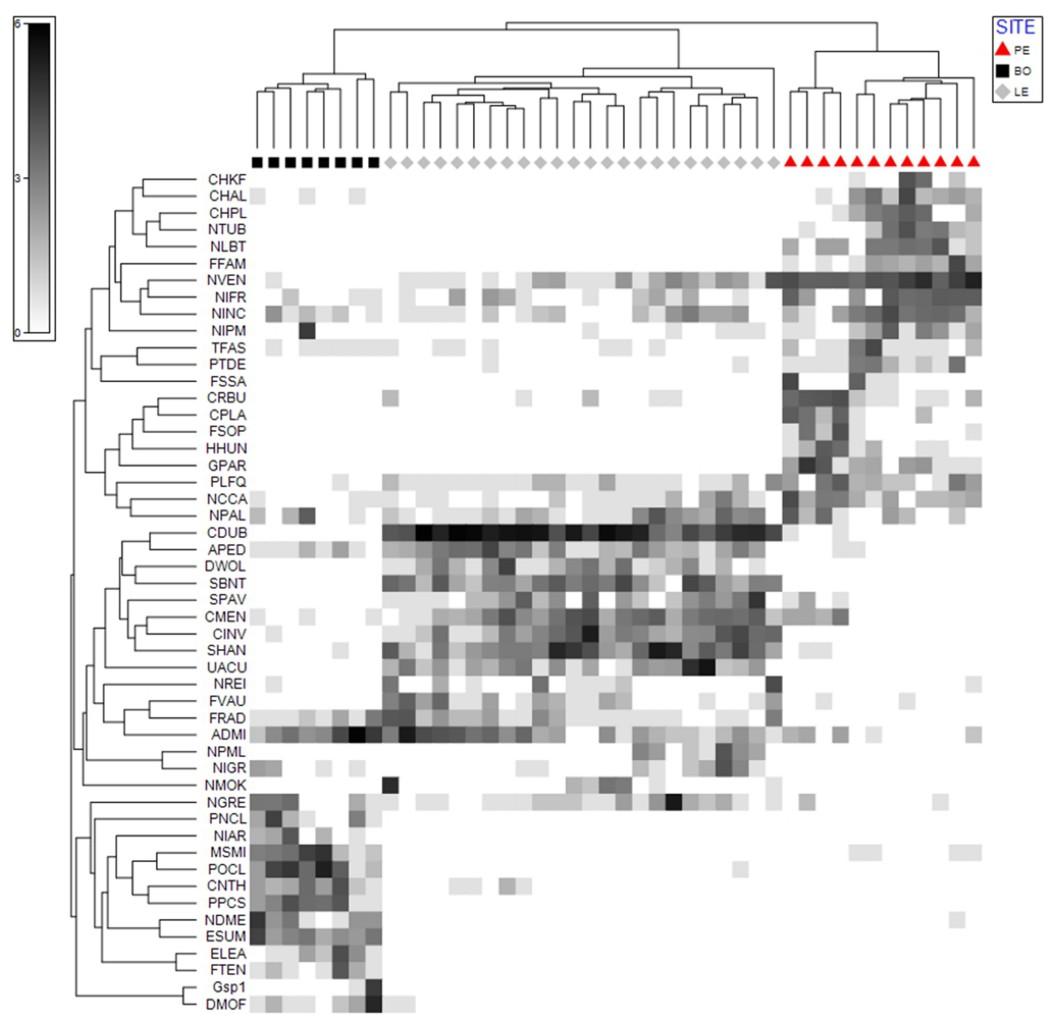

**Figure 3 Shade Plot analysis. The diagram shows the strength of the factor (taxon) affecting the similarity between the samples.** Upper dendrogram—samples divided according to hydrological object. Left dendrogram—50 taxa of diatoms which have the strongest influence on the similarity between the samples.

cloud, with the samples demonstrating high similarity with each other, whereas the samples of D.LEP1, D.LEP2 and D.LEP3 constitute a distinct group, with no clear differentiation into individual reservoirs. In the case of PE, the pond group (D.PEPB) was found to be clearly distinct from the others.

## Shade plot

Shade plot analysis identified 50 species which had the strongest influence on the degree of similarity, or non-similarity, between the samples in the studied ecosystems. Of these taxa, the 11 that most strongly influenced the similarity between the samples in at least two ecosystems were *Navicula veneta* (NVEN), *Navicula cincta* (NCCA), *Navicula gregaria* (NGRE), *Nitzschia frustulum* (NIFR), *Nitzschia inconspicua* (NINC), *Nitzschia palea* (NPAL), *Planothidium frequentissimum* (PLFQ), *Amphora pediculus* (APED), *Cyclotella meneghiniana* (CMEN), *Fragilaria radians* (FRAD) and *Achnanthidium minutissimum* (ADMI) (Fig. 3).

**Table 2 Characteristic species according to SIMPER analysis for studied waterbodies.**

| Pełczyska complex PE | Łęczyca reservoirs LE | Bogdałów reservoir BO |
|---|---|---|
| *Chamaepinnularia krookiformis* (AC) | *Achnanthidium minutissimum* (EC) | *Achnanthidium minutissimum* (EC) |
| *Chamaepinnularia plinskii* (CN) | *Amphora pediculus* (EC) | *Diatoma moniliformis* (EC) |
| *Cocconeis placentula* (CN) | *Cyclostephanos dubius* (EC) | *Encyonopsis subminuta* (EC) |
| *Craticula buderi* (EC) | *Cyclostephanos invisitatus* (EC) | *Mastogloia smithii* (EC) |
| *Craticula halophila* (EC) | *Cyclotella meneghiniana* (EC) | *Nitzschia dissipata* var. *media* (EC) |
| *Fragilaria famelica* (EC) | *Navicula gregaria* (EC) | *Pantocsekiella ocellata* (EC) |
| *Fragilaria sopotensis* (CN) | *Navicula moskalii* (AC) | *Pantocsekiella pseudocomensis* (EC) |
| *Gomphonema parvulum* (EC) | *Nitzschia palea* (EC) | |
| *Hippodonta hungarica* (CN) | *Stephanodiscus binatus* (EC) | |
| *Navicula cincta* (EC) | *Stephanodiscus hantzschii* (EC) | |
| *Navicula veneta* (EC) | *Stephanodiscus parvus* (EC) | |
| *Nitzschia frustulum* (EC) | | |
| *Nitzschia inconspicua* (EC) | | |
| *Nitzschia liebethruthii* (EC) | | |
| *Nitzschia palea* (CN) | | |
| *Nitzschia perminuta* (EC) | | |
| *Nitzschia tubicola* (CN) | | |
| *Planothidium delicatulum* (EC) | | |
| *Planothidium frequentissimum* (EC) | | |
| *Tabularia fasciculata* (EC) | | |

Note:
  EC, euconstant taxa; CN, constant taxa; CN, accessory taxa (*Tümpling & Friedrich, 1999*).

## SIMPER analysis

SIMPER analysis allowed 36 species characteristic of the tested hydrological objects to be distinguished (Table 2). In addition, two species were found to be characteristic of two different ecosystems: *Achnanthidium minutissimum* for LE and BO and *Nitzschia palea* for PE and LE.

## Ecological analysis based on OMNIDIA software

The ecological analysis of diatom assemblages based on data obtained from the OMNIDIA program database, indicated the following:

- pH requirements: while alkaliphilic species predominate in PE (63%), a large percentage in D.PEPB are unknown (24%) or neutrophilic species (23%). The LE reservoirs were dominated by alkalibiontic (45%) and alkaliphilic (24%) organisms. In D.LEP1, 25% of species were unknown. BO was dominated by alkaliphilic (39%) and neutrophilic (29%) species, and 26% of species were unknown (Fig. S2);

- Salinity: the PE complex was characterized by the occurrence of halophilic (43%), oligohalobous (30%) and mesohalobous species (16%); the greatest proportion of the mesohalobous species were found in D.PESB (28%). The LE reservoirs were dominated by

oligohalobous (44%) and halophilic species (42%). BO was dominated by oligohalobous (43%), halophobe (24%) and unknown species (23%) (Fig. S2);

- Nitrogen uptake: the most common species in the PE complex were N-autotrophic tolerant (39%) followed by unknown (25%). The largest percentage of unknown species (28%) was recorded in D.PEPB and D.PEDB. In the LE reservoirs, the most common groups of species were N-autotrophic (57%) and unknown (25%). In BO, 51% species were unknown, 24% were N-autotrophic tolerant and 22% N-autotrophic sensitive (Fig. S2);

- Oxygen requirements: in PE, the largest groups of species were low oxygen (30%), unknown (27%) and moderate oxygen (24%). In LE, oxybiontic species were most common (43%) followed by unknown (25%). In BO, unknown (46%) and polyoxybiontic species (42%) predominated (Fig. S2);

- Sensitivity to saprobity: in PE, the largest group of taxa were α-meso-polysabrobe (28%) and unknown (23%). In D.PEPB, the most abundant was α-meso-polysabrobe (34%) followed by β-mesosaprobe (31%) and unknown (27%). LE primarily included taxa from the α-mesosaprobe group (47%) and unknown (23%). In BO, unknown (34%), β-mesosaprobe (31%) and oligosaprobe taxa (28%) predominated (Fig. S2);

- Trophic status: in PE, the largest group of diatoms were eutrophic (50%) and unknown taxa (25%), LE had the highest percentage (61%) of eutrophic species but also unknown (15%) and hypereutrophic (13%) were present. In BO, the most abundant species were unknown (42%), indifferent (19%) and meso-eutrophic (16%) (Fig. S2);

- Moisture aerophily: in PE, the largest group was aquatic to aerophilic (56%), representing 66% of species in D.PESB, 61% in D.PEDB, and 42% in D.PEPB. The second largest group was unknown (23%), constituting 32% of taxa in D.PEPB. In LE, 37% of the species were aquatic (24% of taxa in D.LEP2), 54% were occasionally aerophilic and 22% were unknown. In BO, the predominant groups of species were unknown (44%) and aquatic to aerophilic (33%) (Fig. S2).

## Characteristic species: OMNIDIA and PCA analysis

The analysis of species characteristic of the tested ecosystems, determined according to *Van Dam, Mertens & Sinkeldam (1994)*, identified 16 taxa classified as 0 in at least one category (Table 3). The next step determined the percentage contribution of each of these species classified as class 0 for the ecological parameters defined by *Van Dam, Mertens & Sinkeldam (1994)* at each sampling point (Table S2).

The PCA was performed to find the relationships between the abiotic parameters and the characteristic species ($n = 36$) (Fig. 4). The Eigenvalues Plot method given eigenvalues above 1%, showed that 12 factors account for 83.2% of the total variance. The first two factors account for 31.3% of the total variance. Based on the PCA analysis for of the 16 characteristic taxa mentioned above and physical and chemical parameters, the following relationships were demonstrated:

- Group A: *Chamaepinnularia krookiformis*, *Chamaepinnularia plinskii*, *Nitzschia liebethruthii* and *Planothidium delicatulum* demonstrate a negative correlation with pH

and a positive correlation with a decrease in the concentrations of $HCO_3^-$, $Ca^{2+}$ and $Fe^{2+/3+}$ (Fig. 4).

- Group B: *Craticula buderi*, *Planothidium frequentissimum* and *Navicula cincta* did not demonstrate any relationship with any water parameters (Fig. 4).

- Group C: *Navicula moskalii*, *Cyclostephanos invisitatus*, *Stephanodiscus parvus*, *S. binatus*, *Diatoma moniliformis*, *Nitzschia dissipata* var. *media*, *M. smithii*, *Pantocsekiella pseudocomensis* and *Encyonopsis subminuta* demonstrated a negative correlation with a decrease in electrolyte conductivity, as well as with the concentrations of $K^+$, $Mg^{2+}$, $Na^+$, $SO_4^{2-}$, $Cl^-$, $PO_4^{3-}$ and $Mn^{3+}$ and water pigments (Fig. 4).

## DISCUSSION

Verification and establishing new ecological indicator values is a key step in standard biomonitoring procedure (*Szczepocka & Żelazna-Wieczorek, 2018*). To specify ecological indicator values or establish new ones we have determined characteristic species. For these species we performed analysis of environmental condition of the ecosystem where they were noted, previous published data and co-occurring species.

### *Planothidium delicatulum* (PTDE) (Figs. 5A–5E)

*Planothidium delicatulum* is a euconstant taxon for PE and an accidental taxon for LE. Its mean percentage share in PE was 2%, and constituted 5% in D.PESB.

*Planothidium delicatulum* does not currently have one ecological indicator value (oxygen requirements) according to *Van Dam, Mertens & Sinkeldam (1994)*.

This species was more abundant in environments such as D.PESB, which was also characterized by the highest concentration of $Cl^-$ (up to 2976 mg $L^{-1}$), elevated electrolytic conductivity, and decreased $K^+$ concentration. The pH of the water in which this species was observed did not exceed 7.

*Planothidium delicatulum* was mainly recorded in salty and brackish environments with neutral or slightly alkaline conditions (*Campeau, Pienitz & Héquette, 1999*; *Gell et al., 2005*; *Caballero et al., 2013*; *Yamamoto, Chiba & Tuji, 2017*; *Van de Vijver, Wetzel & Ector, 2018*).

Based on our findings, we suggest changing the following ecological indicator values in the *Van Dam, Mertens & Sinkeldam (1994)* classification for *Planothidium delicatulum*:

- pH requirements: 3 (neutrophilic) (changing from 5 to 3);
- salinity: 5 (brackish-marine) (changing from 4 to 5).

### *Chamaepinnularia krookiformis* (CHKF) (Figs. 5F–5I) and *Chamaepinnularia plinskii* (CHPL) (Figs. 5J–5M)

In 2016, *Chamaepinnularia krookiformis* was divided into two separate taxa: *Chamaepinnularia krookiformis* and *Chamaepinnularia plinskii* (*Żelazna-Wieczorek & Olszyński, 2016*). Both species were very often recorded together in the same ecosystem. However, the publications which identified *Chamaepinnularia krookiformis* often do not

**Table 3 Characteristics species with classification of ecological indicators values by *Van Dam, Mertens & Sinkeldam (1994)*.**

| Species | Code | Moisture aerophity | Nitrogen uptake | pH requirements | Oxygen requirements | Salinity | Saprobity | Trophic state |
|---|---|---|---|---|---|---|---|---|
| *Achnanthidium minutissimum* | ADMI | 3 | 2 | 3 | 1 | 2 | 2 | 7 |
| *Amphora pediculus* | APED | 3 | 2 | 4 | 2 | 2 | 2 | 5 |
| *Chamaepinnularia krookiformis* | CHKF | 3 | 0 | 3 | 0 | 3 | 1 | 0 |
| *Chamaepinnularia plinskii* | CHPL | 0 | 0 | 0 | 0 | 0 | 0 | 0 |
| *Cocconeis placentula* | CPLA | 2 | 2 | 4 | 3 | 2 | 2 | 5 |
| *Craticula buderi* | CRBU | 0 | 0 | 0 | 0 | 0 | 0 | 0 |
| *Craticula halophila* | CHAL | 2 | 2 | 4 | 2 | 4 | 3 | 5 |
| *Cyclostephanos dubius* | CDUB | 1 | 2 | 5 | 2 | 3 | 3 | 5 |
| *Cyclostephanos invisitatus* | CINV | 0 | 0 | 0 | 0 | 2 | 0 | 5 |
| *Cyclotella meneghiniana* | CMEN | 2 | 3 | 4 | 5 | 3 | 4 | 5 |
| *Diatoma moniliformis* | DMOF | 0 | 0 | 0 | 0 | 0 | 0 | 0 |
| *Encyonopsis subminuta* | ESUM | 0 | 0 | 3 | 1 | 1 | 1 | 1 |
| *Fragilaria famelica* | FFAM | 3 | 1 | 4 | 1 | 2 | 1 | 3 |
| *Fragilaria sopotensis* | FSOP | 1 | 2 | 4 | 1 | 2 | 2 | 4 |
| *Gomphonema parvulum* | GPAR | 3 | 3 | 3 | 4 | 2 | 4 | 5 |
| *Hippodonta hungarica* | HHUN | 3 | 2 | 4 | 3 | 2 | 2 | 4 |
| *Mastogloia smithii* | MSMI | 3 | 0 | 4 | 0 | 4 | 2 | 0 |
| *Navicula cincta* | NCCA | 0 | 0 | 0 | 0 | 2 | 0 | 7 |
| *Navicula gregaria* | NGRE | 3 | 2 | 4 | 4 | 3 | 3 | 5 |
| *Navicula moskalii* | NMOK | 0 | 0 | 0 | 0 | 0 | 0 | 0 |
| *Navicula veneta* | NVEN | 3 | 2 | 4 | 4 | 3 | 4 | 5 |
| *Nitzschia dissipata* var. *media* | NDME | 0 | 0 | 4 | 0 | 2 | 0 | 0 |
| *Nitzschia frustulum* | NIFR | 3 | 4 | 4 | 3 | 3 | 2 | 5 |
| *Nitzschia inconspicua* | NINC | 3 | 3 | 4 | 3 | 3 | 3 | 5 |
| *Nitzschia liebethruthii* | NLBT | 0 | 0 | 5 | 0 | 4 | 0 | 0 |
| *Nitzschia palea* | NPAL | 3 | 4 | 3 | 4 | 2 | 5 | 6 |
| *Nitzschia perminuta* | NIPM | 3 | 1 | 4 | 1 | 2 | 1 | 2 |
| *Nitzschia tubicola* | NTUB | 2 | 3 | 4 | 4 | 3 | 5 | 6 |
| *Pantocsekiella ocellata* | POCL | 1 | 1 | 4 | 1 | 1 | 1 | 4 |
| *Pantocsekiella pseudocomensis* | PPCS | 0 | 0 | 0 | 0 | 0 | 0 | 0 |
| *Planothidium delicatulum* | PTDE | 3 | 1 | 5 | 0 | 4 | 5 | 3 |
| *Planothidium frequentissimum* | PLFQ | 0 | 2 | 4 | 3 | 2 | 4 | 7 |
| *Stephanodiscus binatus* | SBNT | 0 | 0 | 0 | 0 | 0 | 0 | 0 |
| *Stephanodiscus hantzschii* | SHAN | 2 | 3 | 5 | 4 | 2 | 4 | 6 |
| *Stephanodiscus parvus* | SPAV | 0 | 0 | 5 | 0 | 2 | 0 | 6 |
| *Tabularia fasciculata* | TFAS | 3 | 2 | 4 | 3 | 4 | 3 | 5 |

provide appropriate photographic documentation or photos of individual specimens (*Witkowski, 1994*; *Bąk, Witkowski & Lange-Bertalot, 2006*; *Wojtal, 2009*; *Peszek et al., 2015*). Currently available documentation is insufficient to determine whether *Chamaepinnularia krookiformis* and *Chamaepinnularia plinskii* are both present simultaneously in a given environment or whether just one of these species exists.

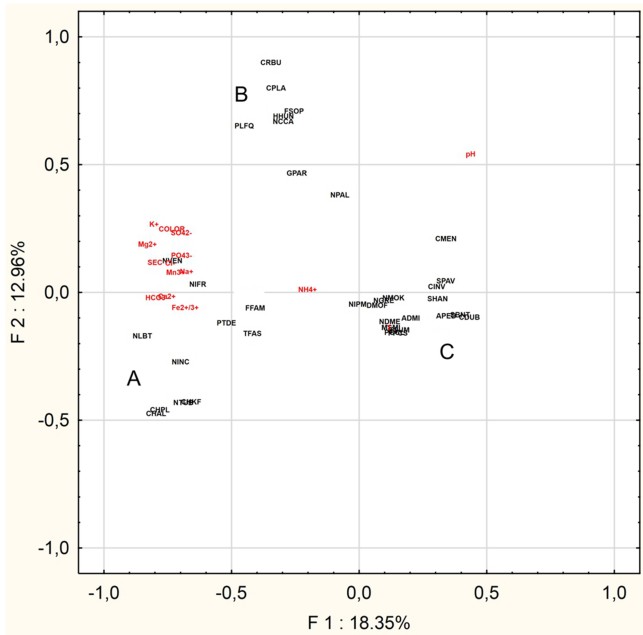

**Figure 4 Principal Component Analysis.** The diagram presents three groups of species A, B and C, whose occurrence can be correlated with selected physical and chemical factors.

*Chamaepinnularia krookiformis* is an accessory taxon for PE (a constant taxon for D.PEDB), *Chamaepinnularia plinskii* is a constant taxon for PE (a euconstant taxon for D.PEDB). The mean share of *Chamaepinnularia krookiformis* was 1.6% in all PE samples, 4% in D.PEDB; for *Chamaepinnularia plinskii*, this amounted to 2.7% in PE, 6% in D.PEDB.

Currently, *Chamaepinnularia krookiformis* lacks three assigned ecological indicator values. For PE, it constitutes 6% of the unknown group in nitrogen uptake, 6% in oxygen requirements and 6% in trophic state (respectively for D.PEDB: 16%, 16% and 15%). *Chamaepinnularia plinskii* has no assigned ecological indicator values and represents 26% of the unknown group for pH requirements, 19% for salinity, 12% for nitrogen uptake, 10% for oxygen requirements, 14% for saprobity, 11% for trophic state and 10% for moisture (respectively for D.PEDB: 60%, 25%, 24%, 23%, 32%, 23% and 28%).

The conditions of the environments in which both species have been recorded indicate that they are class 3 with regard to pH range (neutrophilic). Both species were the most abundant in locations subjected to periodic drying and characterized by high concentrations of chloride ions (up to 1006 mg L$^{-1}$) indicating a brackish environment (*Żelazna-Wieczorek, Olszyński & Nowicka-Krawczyk, 2015*).

On the basis of our findings and those of previous studies (*Krammer & Lange-Bertalot, 1986*; *Krammer, 1992*; *Witkowski, 1994*; *Bąk, Witkowski & Lange-Bertalot, 2006*; *Wojtal, 2009*; *Peszek et al., 2015*; *Żelazna-Wieczorek & Olszyński, 2016*),

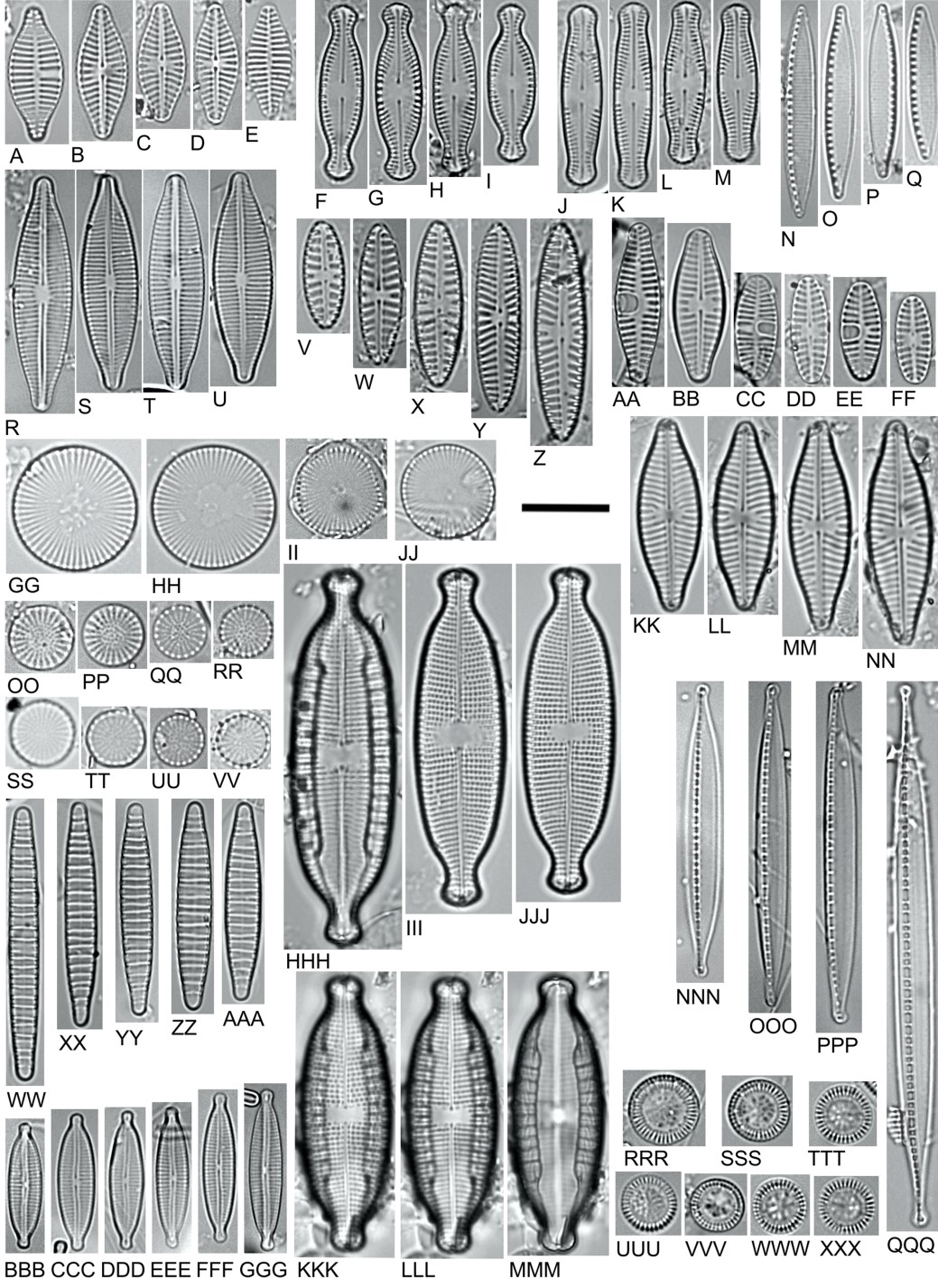

**Figure 5 LM microphotographs of characteristic diatom species.** (A–E). *Planothidium delicatulum.* (F–I) *Chamaepinnularia krookiformis.* (J–M). *Chamaepinnularia plinskii.* (N–Q) *Nitzschia liebethruthii.* (R–U) *Craticula buderi.* (V–Z). *Navicula cincta.* (AA–FF) *Planothidium frequentissimum.* (GG–JJ). *Cyclostephanos invisitatus.* (KK–NN). *Navicula moskalii.* (OO–RR). *Stephanodiscus binatus.* (SS–VV). *S. parvus.* (WW–AAA). *Diatoma moniliformis.* (BBB–GGG). *Encyonopsis subminuta.* (HHH–MMM). *Mastogloia smithii* (KKK–MMM. Same specimen, different focal plane). (NNN–QQQ) *Nitzschia dissipata* var. *media.* (RRR–XXX) *Pantocsekiella pseudocomensis.* Scale bar = 10 μm.

according to *Van Dam, Mertens & Sinkeldam (1994)* ecological indicator system we propose:

   established ecological indicator values for *Chamaepinnularia plinskii*

- pH requirements: 3 (neutrophilic);
- salinity: 4 (mesohalobous);
- trophic state: 5 (eutrophic);
- moisture aerophily: 4 (aerophilic);
- saprobity: 4 (α-meso-polysabrobe);

   for *Chamaepinnularia krookiformis*

- trophic state: 5 (eutrophic);

and the following changes for *Chamaepinnularia krookiformis*

- salinity: from 3 to 4 (mesohalobous);
- moisture aerophily: from 3 to 4 (aerophilic);
- saprobity: from 2 to 4 (β-mesosaprobe to α-meso-polysabrobe).

   Due to the specific conditions and locations of the studied objects, they were exposed to large fluctuations in the inflow of organic matter, mainly from runoff from arable fields and pollution caused by animal grazing. These impurities were manifested as elevated concentrations of $K^+$ ions. Therefore, our results suggest that classifying *Chamaepinnularia krookiformis* as an oligosaprobe is inappropriate. Further tests are needed to determine the optimum occurrence of these species in areas subjected to organic matter loads.

### *Nitzschia liebethruthii* (NLBT) (Figs. 5N–5Q)

*Nitzschia liebethruthii* is a euconstant taxon for PE. Its means percentage share was 4% in the PE samples, and 10% in the D.PEDB samples. It was most numerous in the sample from November 2013 (19%). This species has two specific ecological indicator values. The ecological indicator value analysis for PE found *Nitzschia liebethruthii* to represent 17% of the unknown group in nitrogen uptake, 16% in oxygen requirements, 18% in saprobity, 17% in trophic state and 21% in moisture (respectively for D.PEDB: 37%, 37%, 43%, 36% and 49%).

   *Nitzschia liebethruthii* occurred in environments subjected to periodic drying with a pH close to 7 and high concentration of chloride ions.

   This species was noted in environments with increased salinity, electrolytic conductivity and high pH value (*Rumrich, Lange-Bertalot & Rumrich, 2000*; *Witkowski, Lange-Bertalot & Metzeltin, 2000*; *Lange-Bertalot et al., 2017*; *Földi et al., 2018*).

   We propose established new ecological indicator values according to *Van Dam, Mertens & Sinkeldam (1994)* assigned to *Nitzschia liebethruthii*:

- trophic state: 5 (eutrophic);
- moisture aerophily: 4 (aerophilic);

and following changes:

- pH requirements: from 5 to 3 (alkalibiontic to neutrophilic);
- saprobity: from 2 to 4 (β-mesosaprobe to α-meso-polysabrobe).

### *Craticula buderi* (CRBU) (Figs. 5R–5U)

*Craticula buderi* is a euconstant taxon for PE and an accidental taxon for LE. Its mean percentage share was 4% in all samples for PE, and 12% for D.PEPB. This species has no recorded ecological indicator values. The ecological indicator value analysis for the PE found *Craticula buderi* to constitute 19% of the unknown group in pH requirements, 28% in salinity, 16% in nitrogen uptake, 16% in oxygen requirements, 16% in saprobity, 19% in trophic state and 15% in moisture (respectively for D.PEPB: 53%, 73%, 46%, 46%, 46%, 54% and 41%).

Although *Craticula buderi* was classified into group B (PCA), it was found to be most abundant in environments with an elevated concentration of $Cl^-$ ions, ranging from 685 to 1090 mg $L^{-1}$, (all samples from D.PEPB and one sample from D.PEDB in which the concentration of chloride ions was 1006 mg $L^{-1}$). However, relative abundance was lower in the D.PEPB sample, which was characterized by a chloride ion content of over 1500 mg $L^{-1}$. Interestingly. The concentration of $K^+$ ions exceeded 100 mg $L^{-1}$ at $Cl^-$ concentrations below 1500 mg $L^{-1}$; therefore, it is possible that the decline of this species could be related to the concentration of $K^+$ ions alone. Our observations indicate that the population of *Craticula buderi* from D.PEPB favors a concentration of chloride ions from 500 to 1006 mg $L^{-1}$ which coincides with a $K^+$ ions concentration from 50 to 70 mg $L^{-1}$.

*Craticula buderi* is widespread throughout the world and recognized as cosmopolitan (*Rumrich, Lange-Bertalot & Rumrich, 2000*; *Lange-Bertalot, 2001*; *Bahls, 2009*; *Soltanpour-Gargari, Lodenius & Hinz, 2011*; *Żelazna-Wieczorek, 2011*; *Cichoń, 2016*). This species was found to be dominant in environments characterized by increased electrolytic conductivity and alkaline water (*Holmes & Taylor, 2015*). *Holmes & Taylor (2015)* place *Craticula buderi* in the Bad water quality class. Their recorded values of diatom indices indicate that the environment was eutrophic.

We therefore propose the following classes of ecological indicator values according to *Van Dam, Mertens & Sinkeldam (1994)* for *Craticula buderi*:

- pH requirements: 4 (alkaliphilic);
- trophic state: 5 (eutrophic);
- salinity: 4 (mesohalobous);
- moisture aerophily: 3 (aquatic to aerophilic);

  and following change:

- sabrobity: from 2 to 4 (β-mesosaprobe to α-meso-polysabrobe).

### *Navicula cincta* (NCCA) (Figs. 5V–5Z)

*Navicula cincta* is a euconstant taxon for PE, a constant taxon for LE and an accessory taxon for BO. The mean percentage share of this species for PE is 3%, of which 7% was found in D.PEPB samples.

Currently this species has been assigned two ecological indicator values. The ecological indicator values analysis for the PE found *Navicula cincta* to constitute 24% of the unknown group in pH requirements, 13% in nitrogen uptake, 12% in oxygen requirements, 13% in saprobity, and 12% in moisture (respectively for D.PEPB: 28%, 25%, 25%, 26% and 23%).

An analysis of the physical and chemical data and the variability of occurrence did not show any clear relationships between environmental parameters and the percentage share of *Navicula cincta* in the tested samples. This lack of dependence is also confirmed by the PCA analysis.

*Navicula cincta* has been recorded in various types of ecosystems, although mainly in eutrophic ones with high conductivity. It also tolerates elevated levels of organic matter. This species was also observed in habitats subjected to periodic drying (*Lange-Bertalot & Genkal, 1999*; *Rumrich, Lange-Bertalot & Rumrich, 2000*; *Witkowski, Lange-Bertalot & Metzeltin, 2000*; *Lange-Bertalot, 2001*; *Żelazna-Wieczorek, 2011*; *Wojtal, 2013*; *Lange-Bertalot et al., 2017*). However, several new species from the group *Navicula cincta* s.l. have been described, and it can be assumed that each of these individual species in this group may be associated with narrower optimal ecological conditions (*Cantonati et al., 2016*).

Based on our present findings, and those of previous studies, in our opinion that it is not appropriate to classify *Navicula cincta* as an oligohalobous species with regard to salinity: it has been recorded in fresh (*Żelazna-Wieczorek, 2011*; *Wojtal, 2013*), brackish (*Żelazna-Wieczorek, Olszyński & Nowicka-Krawczyk, 2015*; *Żurek et al., 2018*) and salt waters (*Witkowski, Lange-Bertalot & Metzeltin, 2000*). We propose the following ecological indicator value according to *Van Dam, Mertens & Sinkeldam (1994)* for *Navicula cincta* s.l.:

- moisture aerophily: 3 (aquatic to aerophilic).

Shade Plot analysis found that the presence of *Navicula cincta* s.l. can falsely indicate high similarity between samples from different environments, thus distorting the results of any environmental analysis. Therefore, with regard to the unclear taxonomic status of *Navicula cincta* s.l. and the current lack of knowledge regarding its activities, we recommend this taxon be excluded from the biological assessment of surface water quality.

### *Planothidium frequentissimum* (PLFQ) (Figs. 5AA–5FF)

*Planothidium frequentissimum* is a euconstant taxon for PE and LE. The mean percentage of this species for PE is 2.7%, of which 5% was found in D.PEPB samples. It was most numerous in the D.PEDB.301113 sample (19%). The species has currently six established ecological indicator values. The ecological indicator values analysis for PE found

*Planothidium frequentissimum* to constitute 12% of the unknown group in moisture aerophily (for D.PEPB 14% and D.PESB: 20%).

No relationship was observed between percentage share of *Planothidium frequentissimum* and the changes in chemical and physical parameters in the tested samples. This lack of relationship was confirmed by PCA analysis.

*Planothidium frequentissimum* is an eurytopic species that occurs globally in a variety of habitat types, from natural springs to rivers in urban areas with high levels of pollution. Its value as an indicator is low, as confirmed by the Shade Plot analysis, which found it to significantly affect the degree of similarity observed between samples from different environments (*Siver et al., 2005*; *Levkov et al., 2007*; *Żelazna-Wieczorek, 2011*; *Kulikovskiy, Lange-Bertalot & Kuznetsova, 2015*; *Szczepocka, Nowicka-Krawczyk & Kruk, 2018*).

Recently *Planothidium frequentissimum* was divided into several different species. In studied samples we observe several species which belong to *Planothidium frequentissimum* s.l. (Figs. 5AA–5FF) (*Wetzel et al., 2019*). A light microscope (LM) is still used to identify species in ecological research and biological assessment of aquatic ecosystems. Due to the likeness of the basic morphological features of the newly described species observed in LM, especially in the case of *Planothidium frequentissimum* s.s. and *Planothidium straubianum*, distinguishing them will be difficult or limited, which may lead to errors in the assessment. We therefore recommend that *Planothidium frequentissimum* s.l. be excluded from the biological assessment of surface water quality.

### *Cyclostephanos invisitatus* (CINV) (Figs. 5GG–5JJ)

*Cyclostephanos invisitatus* is a euconstant taxon for LE. Its percentage share for LE was 4.8%. *Cyclostephanos invisitatus* currently has two ecological indicator values assigned. The ecological indicator values analysis for LE found it to constitute 24% of the unknown group in pH requirements, 17% in nitrogen uptake, 18% in oxygen requirements, 19% in saprobity, and 20% in moisture.

*Cyclostephanos invisitatus* occurs in diverse environments, however, it is most frequently reported in aquatic ecosystems subjected to high human impact, alkaline and high conductivity (*Reavie & Smol, 1998*; *Yang et al., 2005*; *Wojtal & Kwandrans, 2006*; *Kiss et al., 2012*; *Houk, Klee & Tanaka, 2014*; *Reavie & Kireta, 2015*; *Olszyński & Żelazna-Wieczorek, 2018*).

We therefore propose that the following classes of ecological indicator values according to *Van Dam, Mertens & Sinkeldam (1994)* be established for *Cyclostephanos invisitatus*:

- pH requirements: 4 (alkaliphilic);
- moisture aerophily: 1 (aquatic).

### *Navicula moskalii* (NMOK) (Figs. 5KK–5NN)

*Navicula moskalii* is an accessory taxon for LE. Its mean percentage share for LE was 1.5%. Its incidence was greatest in sample D.LEP1.250315 (26%).

*Navicula moskalii* has no assigned ecological indicator values. The ecological indicator value analysis for LE found it to constitute 5% of the unknown group in pH requirements,

7% in salinity, 5% in nitrogen uptake, 5% in oxygen requirements, 5% in saprobity, 6% in trophic state and 5% in moisture. The greatest occurrence of *Navicula moskalii* was observed in samples with the highest concentrations of $Ca^{2+}$ (143.6 mg $L^{-1}$), $HCO_3^-$ (338.6 mg $L^{-1}$), $SO_4^{2-}$ (146.9 mg $L^{-1}$) and with high $Mg^{2+}$ content.

*Navicula moskalii* was observed in a number of ecosystems (*Metzeltin & Witkowski, 1996*; *Lange-Bertalot, 2001*; *Żelazna-Wieczorek, 2011*; *Noga et al., 2016*; *Lange-Bertalot et al., 2017*), particularly in eutrophic waters with an elevated level of $Ca^{2+}$ and $HCO_3^-$ ions. *Żelazna-Wieczorek (2011)* report a significant number of *Navicula moskalii* in springs with high levels of eutrophication, however with $Ca^{2+}$, $SO_4^{2-}$, $HCO_3^-$ and $Mg^{2+}$ concentrations lower than those in the LE samples.

We therefore propose that the following classes of ecological indicator values according to *Van Dam, Mertens & Sinkeldam (1994)* be established for *Navicula moskalii*:

- pH requirements: 4 (alkaliphilic);
- salinity: 2 (oligohalobous);
- trophic state: 7 (indifferent).

### *Stephanodiscus binatus* (SBNT) (Figs. 5OO–5RR)

*Stephanodiscus binatus* is a euconstant taxon for LE. Its percentage share for LE was 4.3%. *S. binatus* has no recorded ecological indicator values. The ecological indicator value analysis for LE found it to constitute 25% of the unknown group in pH requirements, 47% in salinity, 18% in nitrogen uptake, 18% in oxygen requirements, 20% in saprobity, 29% in trophic state and 21% in moisture.

The largest percentage share of *S. binatus* was recorded in the spring months and the lowest in autumn. Its abundance was found to be elevated in December 2014 and 2015; the same samples demonstrated the highest concentrations of $Ca^{2+}$, $Mg^{2+}$ and the highest pH (above 8).

*Stephanodiscus binatus* has been recorded in various water ecosystems ranging from oligotrophic to eutrophic; however, all are characterized by elevated pH value (*Stoermer & Håkansson, 1984*; *Håkansson & Kling, 1990*; *Houk, Klee & Tanaka, 2014*; *Olszyński & Żelazna-Wieczorek, 2018*).

We therefore propose that the following classes of ecological indicator values according to *Van Dam, Mertens & Sinkeldam (1994)* be established for *S. binatus*:

- pH requirements: 4 (alkaliphilic);
- salinity: 2 (oligohalobous);

### *Stephanodiscus parvus* (SPAV) (Figs. 5SS–5VV)

*Stephanodiscus parvus* is a euconstant taxon for LE and an accidental taxon for PE. Its percentage share for LE was 2.4%. It was most abundant in the D.LEP3.260714 sample (22%). This species has three assigned ecological indicator values. The ecological indicator values analysis for LE found the taxon to constitute 9% of the unknown group in nitrogen uptake, 9% in oxygen requirements, 10% in saprobity, and 10% in moisture.

Stephanodiscus parvus is noted mainly in eutrophic hypereutrophic ecosystems with elevated electrolytic conductivity. It is also a good indicator of waters with a strong anthropogenic impact (*Reavie & Smol, 1998*; *Reavie & Kireta, 2015*; *Olszyński & Żelazna-Wieczorek, 2018*; *Reavie & Cai, 2019*).

Based on our findings and literature data, we propose the following change in ecological indicator values according to *Van Dam, Mertens & Sinkeldam (1994)* for *S. parvus*:

- pH requirements: from 5 to 4 (alkalibiontic to alkaliphilic).

### *Diatoma moniliformis* (DMOF) (Figs. 5WW–5AAA)

*Diatoma moniliformis* is a euconstant taxon for BO. Its mean percentage share for BO was 3.9%. It currently has no assigned ecological indicator values. According to the ecological indicator values analysis for BO, this taxon constituted 10% of the unknown group in pH requirements, 11% in salinity, 8% in nitrogen uptake, 8% in oxygen requirements, 9% in saprobity, 9% in trophic state and 8% in moisture.

*Diatoma moniliformis* was found in 87.5% of samples from BO. Interestingly, it constituted 28% of the share in one sample from December 2016 (D.BOZB.091216); however, its share was below 2% in the previous season, and was not higher than 1–2% in the other samples from December 2016. The chemical and physical characteristics of D.BOZB.091216 did not differ significantly from those of the other samples.

This species is also found in fresh and salt water, as well as the Baltic and arctic areas with high conductivity (*Potapova & Snoeijs, 1997*; *Rumrich, Lange-Bertalot & Rumrich, 2000*; *Levkov et al., 2007*; *Pniewski & Sylwestrzak, 2018*).

One of the factors that influences the abundance of *D. moniliformis* is the water temperature. Studies indicate that temperatures above 10–15 °C (*Potapova & Snoeijs, 1997*; *Pniewski & Sylwestrzak, 2018*) are associated with population growth. However, populations have been observed in freshwater streams and lakes in arctic areas, in which the temperature of the water is below 10 °C (*Antoniades, Douglas & Smol, 2005*). Population growth was also observed at 4.7 °C in sample D.BOZB.091216; therefore, low temperature may have an influence on the abundance of this species.

### *Encyonopsis subminuta* (ESUM) (Figs. 5BBB–5GGG)

*Encyonopsis subminuta* is a euconstant taxon in BO, where its mean percentage share was 4.1%. Presently, *E. subminuta* has been assigned five ecological indicator values. Ecological indicator values analysis for BO found it to constitute 7% of the unknown group in nitrogen uptake and 8% in moisture.

*Encyonopsis subminuta* was found to be most abundant in sample D.BOZB.041115. The sample was also characterized by an elevated concentration of $Fe^{2+/3+}$ ions and the lowest pH value. In subsequent samples, when the concentration of Fe ions dropped, the abundance of *E. subminuta* also decreased.

*Encyonopsis subminuta* is regarded as a cosmopolitan taxon, occurring in the temperate and boreal zone. It is most abundant in oligo- to mesotrophic waters with

electrolytic conductivity between 190 and 250 µS cm$^{-1}$ (*Krammer, 1997*; *Noga et al., 2014*; *Novais et al., 2014*; *Feret, Bouchez & Rimet, 2017*).

*Encyonopsis subminuta* may be sensitive to the concentration of Fe ions; however, the increase of these ions is associated with a drop in pH. Our research confirms that the optimal pH for population size is close to 7.

### *Mastogloia smithii* (MSMI) (Figs. 5HHH–5MMM)

*Mastogloia smithii* is a euconstant taxon for BO. Its mean percentage share for BO was 6.3%. it was found in greatest numbers in D.BOZB.300615 (22%) and D.BOZB.261016 (15%). *M. smithii* has been assigned four ecological indicator values. The ecological indicator values analysis for the BO found this species to constitute 13% of the unknown group in nitrogen uptake, 14% in oxygen requirements and 15% in trophic state.

The environment in BO regarding salinity was classified according to *Van Dam, Mertens & Sinkeldam (1994)* as oligohalobus (43% species); however, 7% of the mesohalobous species were represented by one species: *M. smithii*.

This species is recorded in fresh, brackish and salt water (*Witkowski, Lange-Bertalot & Metzeltin, 2000*; *Busse & Snoeijs, 2003*; *Weckström & Juggins, 2005*; *Martinez-Goss & Evangelista, 2011*; *Lange-Bertalot et al., 2017*). Its presence in environments with varying degrees of salinity may suggest that this does not have an significant influence on population size.

Based on our present findings and literature data, we propose the following change in the ecological indicator values according to *Van Dam, Mertens & Sinkeldam (1994)* for *M. smithii*:

- salinity: from 4 to 3 (mesohalobous to halophilic).

### *Nitzschia dissipata* var. *media* (NDME) (Figs. 5NNN–5QQQ)

*Nitzschia dissipata* var. *media* is a euconstant taxon for BO. Its mean percentage share for BO was 3.8%, and the highest proportion (20%) was found in D.BOZB.041115. *Nitzschia dissipata* var. *media* has been assigned two ecological indicator values. Ecological indicator values analysis for BO found it to constitute 9% of the unknown group in nitrogen uptake, 10% in oxygen requirements, 11% in saprobity, 11% in trophic state and 9% in moisture.

The increase in occurrence of *Nitzschia dissipata* var. *media* is associated with an increase in the level of Fe$^{2+/3+}$, similar to *Encyonopsis subminuta*. In addition, it was found in the ecosystem, that is, BO, with the lowest concentrations of ions indicative of the presence of organic pollutants in the environment, such as K$^+$ and NH$_4^+$.

Although *Nitzschia dissipata* var. *media* is found sporadically, it is commonly found in oligo- to mesotrophic waters with a pH between 7 and 8 (*Krammer & Lange-Bertalot, 1997*; *Van de Vijver, Frenot & Beyens, 2002*; *Antoniades, Douglas & Smol, 2005*; *Żelazna-Wieczorek, 2011*; *Lange-Bertalot et al., 2017*).

**Table 4 Selected 16 characteristic species with the new or altered (bold) ecological indicator values according to *Van Dam, Mertens & Sinkeldam (1994)*.**

| Species | Code | Moisture aerophity | Nitrogen uptake | pH requirements | Oxygen requirements | Salinity | Saprobity | Trophic state |
|---|---|---|---|---|---|---|---|---|
| *Chamaepinnularia krookiformis* | CHKF | **4** | 0 | 3 | 0 | **4** | **4** | **5** |
| *Chamaepinnularia plinskii* | CHPL | **4** | 0 | **3** | 0 | **4** | **4** | **5** |
| *Craticula buderi* | CRBU | **3** | 0 | **4** | 0 | **4** | **4** | **5** |
| *Cyclostephanos invisitatus* | CINV | **1** | 0 | **4** | 0 | 2 | 0 | 5 |
| *Diatoma moniliformis* | DMOF | 0 | 0 | 0 | 0 | 0 | 0 | 0 |
| *Encyonopsis subminuta* | ESUM | 0 | 0 | 3 | 1 | 1 | 1 | 1 |
| *Mastogloia smithii* | MSMI | 3 | 0 | 4 | 0 | **3** | 2 | 0 |
| *Navicula cincta* | NCCA | **3** | 0 | 0 | 0 | 2 | 0 | 7 |
| *Navicula moskalii* | NMOK | 0 | 0 | **4** | 0 | **2** | 0 | **7** |
| *Nitzschia dissipata* var. *media* | NDME | 0 | 0 | 4 | 0 | 2 | **2** | 0 |
| *Nitzschia liebethruthii* | NLBT | **4** | 0 | **3** | 0 | 4 | **4** | **5** |
| *Pantocsekiella pseudocomensis* | PPCS | 0 | 0 | 0 | 0 | 0 | 0 | 0 |
| *Planothidium delicatulum* | PTDE | 3 | 1 | **3** | 0 | **5** | 5 | 3 |
| *Planothidium frequentissimum* | PLFQ | 0 | 2 | 4 | 3 | 2 | 4 | 7 |
| *Stephanodiscus binatus* | SBNT | 0 | 0 | **4** | 0 | **2** | 0 | 0 |
| *Stephanodiscus parvus* | SPAV | 0 | 0 | **4** | 0 | 2 | 0 | 6 |

Based on our findings and literature data, we propose the following ecological indicator values according to *Van Dam, Mertens & Sinkeldam (1994)* for *Nitzschia dissipata* var. *media*:

- saprobity: 2 (β-mesosaprobe)

### *Pantocsekiella pseudocomensis* (PPCS) (Figs. 5RRR–5XXX)

*Pantocsekiella pseudocomensis* is a euconstant taxon for BO. Its mean percentage share for BO was 4.4%. It was most abundant in D.BOZB.250315 (9%) and in D.BOZB.220616 (10%). *Pantocsekiella pseudocomensis* has not been assigned any ecological indicator values according to *Van Dam, Mertens & Sinkeldam (1994)*. The ecological indicator values analysis for BO found it to represent 22% of the unknown group in pH requirements, 25% in salinity, 10% in nitrogen uptake, 11% in oxygen requirements, 18% in saprobity, 12% in trophic state and 13% in moisture.

The greatest amount of *Pantocsekiella pseudocomensis* was found in samples characterized by the highest levels of ammonium ions. Its percentage share was lowest in samples with the lowest water temperature, apart from D.BOZB.250315.

Currently, *Pantocsekiella pseudocomensis* is assigned to the *Pantocsekiella comensis* complex, with *Pantocsekiella comensis* and *Pantocsekiella costei*. In our opinion that assigning ecological indicator values for particular species of the *Pantocsekiella comensis* complex is unjustified at the current state of knowledge, and that all species within the complex should be assigned the same provisional ecological indicator values until their individual properties are better understood (*Houk, Klee & Tanaka, 2010*; *Kistenich et al., 2014*; *Duleba et al., 2015*).

## CONCLUSIONS

The water ecosystems created in the post-mining areas create a complex of conditions that are not found in other natural ecosystems, and the benthic diatom species present in such environments are very often present in higher numbers than in other habitats. The specific hydro-geological conditions prevailing in the post-production reservoirs provide a unique opportunity to observe interspecies differences and intra-species variability, allowing for the verification or isolation of new taxa and a greater insight into their autecology (*Żelazna-Wieczorek & Olszyński, 2016*; *Olszyński & Żelazna-Wieczorek, 2018*).

The identification of species characteristic of the studied ecosystems may foster further growth of ecological research and increase the reliability of surface water quality assessment, as such knowledge is needed to verify their ecological indicator values, and hence calculate diatoms indices with greater accuracy.

Ecological indicator values as set out by *Van Dam, Mertens & Sinkeldam (1994)* are utilized in many ecological works describing the ecological conditions of the studied ecosystems. These ecological indicator values form the basis for calculating diatoms indices describing the ecological state of surface waters. It is therefore necessary to constantly update and establish new ecological indicator values for particular diatom species.

Many authors who describe new species, or encounter existing species in new ecosystems, regularly propose updates for individual ecological indicator values. However, these findings, may not be introduced and updated in the OMNIDIA program for a number of years. The OMNIDIA system is used by state institutions in many countries around the world to assess surface water quality (*Campeau, Pienitz & Héquette, 1999*; *Rumrich, Lange-Bertalot & Rumrich, 2000*; *Witkowski, Lange-Bertalot & Metzeltin, 2000*; *Gell et al., 2005*; *Potapova & Ponader, 2008*; *Wojtal & Sobczyk, 2012*; *Caballero et al., 2013*; *Żelazna-Wieczorek & Olszyński, 2016*; *Yamamoto, Chiba & Tuji, 2017*; *Lange-Bertalot et al., 2017*; *Földi et al., 2018*; *Van de Vijver, Wetzel & Ector, 2018*). Clearly, if these assessments are based on incomplete or outdated data, assessments of aquatic environments may be fraught with error.

The present study used three principles to identify proposed changes in the classification of ecological indicator values for characteristic species according to the *Van Dam, Mertens & Sinkeldam (1994)* system, or to establish new values which were previously absent: the analysis of environmental conditions prevailing in the studied ecosystems, the analysis of relevant literature data, and references to the ecological indicator values of other species (Table 4). This mode of research can serve as a model for updating databases used to assess surface water quality.

### Funding

The authors received no funding for this work.

## Competing Interests

The authors declare that they have no competing interests.

## Author Contributions

- Rafał M. Olszyński performed the experiments, analyzed the data, prepared figures and/or tables, authored or reviewed drafts of the paper, approved the final draft.
- Ewelina Szczepocka analyzed the data, authored or reviewed drafts of the paper, approved the final draft.
- Joanna Żelazna-Wieczorek conceived and designed the experiments, analyzed the data, contributed reagents/materials/analysis tools, authored or reviewed drafts of the paper, approved the final draft.

## Data Availability

Raw data is available in the Supplemental Files.

## Supplemental Information

Supplemental information for this article can be found online at http://dx.doi.org/10.7717/peerj.8117#supplemental-information.

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
