# Peer review of "Critical multi-stranded approach for determining the ecological values of diatoms in unique aquatic ecosystems of anthropogenic origin"

_PeerJ, doi:10.7717/peerj.8117_

## Round 0.1 · original submission · Major Revisions

The reviewers have commented on your above paper. They indicated that it is not acceptable for publication in its present form.
However, if you feel that you can suitably address the reviewers' comments (included), I invite you to revise and resubmit your manuscript. Please, see also the annotated manuscript.

Reviewer 1 ·

Basic reporting

English is competent enough, but requires some polishing by the authors, and maybe a quick look by an external expert with excellent knowledge of the both technical English and diatom ecology. In general paper is written competently, though some of the sentence i marked can be made more succinct, especially huge list of the species (lines 250-263), which should be made into the table.
Literature referencing seems to be sufficient, and raw data are provided, but I would argue that results of the paper are currently irreproducible and this is tied to the soft being used in statistical analysis, OMNIDIA package in particular. This is THE MOST important issue of the manuscript. While I do understand that "Lecointe C., Coste M. & Prygiel J. 1993. “Omnidia:” a software for taxonomy, calculation of
diatom indices and inventories management. Hydrobiologia 269/270: 509-513." is a standard reference and software piece for the analysis in the Diatom ecology, reporting standards of the original paper describing the software is substandard for 2019. It is closed code without any data on the implementation of the ecological indices included. I would suggest that authors should do complete "dissection" of the math and stat assumptions behind the package, maybe as a supplement. If it is impossible to do so, due to the commercial nature of the programm, you should check the validity of the findings using alternative statistical approaches. Most importantly, I looked OMNIDIA up https://omnidia.fr/presentation/ and programm does not seems to provide any sort of the statistical power of the test measure. My suggestion, back up your fundings with RDA analysis to see which abiotic factors influencing the compositions of your communities and GLM type linear models to see which factors where most important for the every species of the diatoms. I would like authors to provide the data about underlying assumptions on the data, model output and indicators of the size of the difference/ magnitude of the effects - preferably effect size, p-values could be used but effect size is preferable.

Figures are fine, but many other stat test are required before amount of illustrative material will be sufficient to support the claims of the paper.
Paper is not self contained until statistical procedures are better explained.

Experimental design

Research is original and within the scope of the journal.
Sampling design is fine, but more details are required on the abiotic factors measurements.
Authors undertook admirable work and processed impressive amount of the data, but results are currently irreproducible, see above.

Validity of the findings

I can not access the validity of the findings until I see more statistical tests and explanation of the methods.

Annotated reviews are not available for download in order to protect the identity of reviewers who chose to remain anonymous.

Reviewer 2 ·

Basic reporting

 Comment on language and grammar issues
In general the manuscript (ms) is well written, although it would benefit from a general revision of the English language.
I enclose 2 examples of required improvement of the English language:
- sentence of lines 71 to 73
- sentence lines 102-103

 Intro & background to show context
Introduction and background include the main issues that are reported along the manuscript.

 Literature well referenced & relevant
All references cited in the text are listed in the final reference list.
All references in the final reference list are cited in the text.
I didn’t verify the reference formatting for this journal, nevertheless, I found some differences in formatting between references in the final list (i.e.: separation between authors’ names with a comma, and no type of separation see Bak et al. 2006 and Campeau et al 1999).

 Figures are relevant, high quality, well labelled & described
All figures and tables are useful and of good quality.
A few suggestions were added in the captions of the pdf of the ms.
 Raw data supplied
I didn´t find a caption for the excel table containing the raw data. It would be useful to have a caption explaining what the numbers in the species list mean (probably number of valves counted?). The + sign should also be explained. I would recommend the insertion of the authorities of the diatom taxa in the excel table, so that at least once along the manuscript it will be possible to read this information.
Why are some environmental parameters missing? In the ms it was only said that 44 samples were taken and that diatoms and water for chemical analysis were performed.
The comma (,) separating the numbers of the environmental parameters should be replaced by and end stop (.) as decimal places should be represented by end stop and not a comma.

Experimental design

 Original primary research
The research on diatom ecology is not new. As the authors refer many diatom indices are based on the knowledge about diatom ecology, so in fact the research presented in the ms refers to the improvement/definition of ecological values for some environmental parameters and for 3 extreme environments which were previously mining areas. This primary research did not exist before for these areas, and therefore it is useful new data for scientific community working on diatoms.
 Research question well defined, relevant & meaningful. It is stated how the research fills an identified knowledge gap
Yes
 Rigorous investigation performed to a high technical & ethical standard
The authors performed good field sampling design based on previous knowledge about mining activities, field sampling and laboratorial work seems to have been rigorously performed as well, at least at the diatom level as the raw data provided allow to .confirm that authors were careful with identification and counting of samples. The photomicrographs of diatoms presented are of good quality and diatoms have been correctly identified. Nevertheless, the results extracted from the obtained field and lab data were not extensively explored as I refer several times along the ms in the comments in the pdf. For example: the redefinition or proposal of ecological values for some diatoms are not fully explained or justified through mathematical analysis (objective) but are rather more or less attributed in a subjective way with only verbal definition. The inclusion of available information in the literature on these taxa ecological values is interesting and important and helps to summarize existing autecological data.
 Methods described with sufficient detail & information to replicate
Yes, although I included in the ms the need to clarify some issues about sampling diatoms, i.e.: what kind of sampling was done (type of substrate for benthic diatoms, phytoplankton net for planktonic diatoms, any other habitat?). The list of diatoms presented show that benthic habitats must have been sampled as the majority of taxa are benthic and not planktonic. Another aspect to clarify and related to the previous comment is the clarification about the type of waterbodies sampled i.e.: are they all lentic in nature (reservoirs) or are there some lotic ones as well (streams).
The methods for chemical analysis were not included. This may be replaced by the reference of the manual with the methods used (for example APHA).

Validity of the findings

 Impact and novelty not assessed. Negative/inconclusive results accepted. Meaningful replication encouraged where rationale & benefit to literature is clearly stated.
The authors were careful in explaining the novelty and benefits of their research, with which I agree.
 All underlying data have been provided; they are robust, statistically sound, & controlled.
Yes
 Speculation is welcome, but should be identified as such.
I don’t consider that the redefinition and proposal of ecological values, despite the lack of full explanation, are not speculative.
 Conclusions are well stated, linked to original research question & limited to supporting results.
Yes

Additional comments

Authors use the expression “ecological indicator values” throughout the ms, and in fact it is the main focus of the research. I understand that authors use this expression as it is referred in the OMNIDIA software under Van Dam’s ecological results. Nevertheless, I would like to point out that most autoecological diatom indices are based on 3 parameters: average relative abundance of taxa, the sensibility value (provides information about the response of taxa to degradation and usually varies between 1 and 5) and indicator value (provides information about the degree of stenoicity for each taxon, i.e. measures the reliability of each taxon for a certain environment). So, what I think authors are referring to are not “ecological indicator values” but the sensibility values. I therefore recommend authors to remove the word “indicator” from the expression they use throughout the entire manuscript maintaining as “ecological values” only.
The establishment of new or modified ecological values for the diatom species in this study should be better explained and related to abiotic values and ranges of values. Authors should justify why a certain taxon is changed from 1 class to another in Van Dam’s classification based on the range of chemical parameters. What does eutrophic condition mean? To be eutrophic what concentration of nitrogen and phosphorus must be present in the water?
In order to make the attribution/modification of ecological values more objective and understandable authors should correlate taxa abundances with environmental parameters. It seems that these changes are proposed but in a somewhat subjective way as it is mainly based on the PCA analysis and literature.
The environmental range reported in the manuscript for the set of sampling sites is not very wide and includes sites with very specific and particular characteristics, therefore I advise caution in the attribution of ecological values as these only refer to a small range of environmental conditions.
This manuscript contains interesting information and I recommend it for publication, but after the changes suggested.

Annotated reviews are not available for download in order to protect the identity of reviewers who chose to remain anonymous.

Reviewer 3 ·

Basic reporting

See Attached PDF

Experimental design

See Attached PDF

Validity of the findings

See Attached PDF

Annotated reviews are not available for download in order to protect the identity of reviewers who chose to remain anonymous.

---

## Round 0.2 · Minor Revisions

Your manuscript should be improved regarding the inclusion of some genus name (see comments for the author from Rev 3).

Reviewer 3 ·

Basic reporting

I recommend an acceptance of the manuscript for publication in your journal. The comments on the improvements in the first review have been made to the best possible satisfaction. In the resubmission of the revised manuscript I see no need for further improvement. The authors have revised the manuscript in an acceptable manner and thus contributed to the increase in quality.

Experimental design

The improvements have been made to the best possible satisfaction.

Validity of the findings

All underlying data have been provided.

Additional comments

I recommend an acceptance of the manuscript for publication in your journal. The comments on the improvements in the first review have been made to the best possible satisfaction. In the resubmission of the revised manuscript I see no need for further improvement. The authors have revised the manuscript in an acceptable manner and thus contributed to the increase in quality.

For improvement:
- at the beginning of the sentence, please enter the whole genus name for C. invisitatus - line 503, S. binatus lines 535-536, S. parvus – line 557, D. moniliformis – line 569, E. subminuta – line 593, M. smithii – lines 601-602.

---

## Round 0.3 · accepted · Accept

Thank you again for improving your manuscript and for choosing our Journal.